# Spectrum Sharing in the Sky and Space: A Survey

**DOI:** 10.3390/s23010342

**Published:** 2022-12-29

**Authors:** Ling Zhang, Zhiqing Wei, Lin Wang, Xin Yuan, Huici Wu, Wenyan Xu

**Affiliations:** 1Key Laboratory of Universal Wireless Communications, Ministry of Education, School of Information and Communication Engineering, Beijing University of Posts and Telecommunications, Beijing 100876, China; 2Data61, Commonwealth Scientific and Industrial Research Organisation (CSIRO), Marsfield, NSW 2122, Australia; 3National Engineering Lab for Mobile Network Technologies, Beijing University of Posts and Telecommunications, Beijing 100876, China

**Keywords:** space–air–ground integrated network, spectrum sharing, cognitive radio, spectrum sensing, interference management, survey, review

## Abstract

In order to achieve the vision of seamless wireless communication coverage, a space–air–ground integrated network is proposed as a key component of the sixth-generation (6G) mobile communication system. However, the spectrum used by aerial networks has become gradually crowded with the increase in wireless devices. Space networks are also in dire need of developing new bands to address spectrum shortages. As an effective way to solve the spectrum shortage problem, spectrum sharing between aerial/space networks and ground networks has been extensively studied. This article summarizes state-of-the-art studies on spectrum sharing between aerial/space networks and ground networks. First, this article provides an overview of aerial networks and space networks and introduces the main application scenarios of aerial networks and space networks. Then, this article summarizes the spectrum sharing techniques between aerial/space networks and ground networks, including existing spectrum utilization rules, spectrum sharing modes and key technologies. Finally, we summarize the challenges of spectrum sharing between aerial/space networks and ground networks. This article provides guidance for spectrum allocation and spectrum sharing of space–air–ground integrated networks.

## 1. Introduction

With the rapid development of technologies such as the Internet of Things (IoT) and Industrial Internet, the demands for Human-to-Human (H2H), Human-to-Machine (H2M), Machine-to-Machine (M2M) and Machine in/or Humans (MiH) communications have increased dramatically [1], and devices connected to the Internet are showing an explosive growth [2]. The increase in communication demand and the sharp increase in equipment have brought about the rapid growth of spectrum demand, and they have also brought severe challenges to the further development of ground networks [3]. Additionally, only using ground networks cannot provide communication services at any time or in any region. In terms of coverage and ability to withstand natural disasters, etc., only using ground base stations (BSs) still has certain limitations. For example, remote areas are sparsely populated, and the cost of constructing ground BSs is relatively high. In areas with many obstacles, such as mountainous areas, the quality of communication services provided by ground networks cannot be guaranteed. In addition, ground BSs are relatively weak against natural disasters. In the event of a severe natural disaster, ground BSs are easily destroyed, resulting in paralysis of communication services [4]. For example, in August 2021, heavy rain caused flooding in some areas of the Henan Province in China. Ground communication facilities suffered great damage during this heavy rain, which hampered the rescue mission.

Due to the advantages of wide coverage and the ability to deploy at a high altitude, aerial networks and space networks can compensate for the shortcomings of current ground networks. Compared with ground networks, aerial networks and space networks have a wider coverage and are expected to improve communication performance in remote areas and maritime areas. For example, SpaceX plans to launch nearly 120,000 satellites through its Starlink project, aiming to provide wireless communication services, as well as establish a satellite communication network [5]. In recent years, space network infrastructure (such as satellites) and aerial network infrastructure (such as unmanned aerial vehicles (UAVs), aircrafts, etc.) have also been used in emergency communications to quickly restore communication services in disaster-stricken areas. In August 2017, China Mobile deployed an aerial BS (i.e., a UAV) in Jiuzhaigou, where an earthquake occurred. The aerial BS restored a network coverage of over 30 square kilometers, providing a powerful guarantee of rapid communication signal recovery.

To meet higher communication requirements, 6G proposes the vision of a space–air–ground integrated network, aiming to achieve seamless global coverage of communication networks [6]. However, with the rapid growth of wireless communication equipment, the current spectrum is scarce. Aerial network devices, such as UAVs and aircrafts, mainly provide communication services by accessing unlicensed frequency bands (2.4 GHz and 5 GHz). There are multiple devices on unauthorized frequency bands, such as IOT devices [7]. With an increasing number of wireless devices accessing unlicensed frequency bands, serious interference between devices is likely to occur. In addition, with the large-scale application of aerial BSs in the future, these frequency bands cannot meet higher communication requirements. For space networks, although dedicated bands (L-band, S-band, etc.) are currently allocated, it is urgent to develop new frequency bands to provide communication services in the face of rapidly increasing equipment. To address the aforementioned spectrum shortage, cognitive radio (CR) technology has been extensively studied. Different from the static spectrum allocation strategy, CR technology aims to improve spectrum utilization. By using methods such as controlling the transmitter power and beam angle, multiple networks can access the spectrum simultaneously. When ground networks are paralyzed by natural disasters, aerial networks or space networks can detect the idle spectrum of ground networks in this period to realize the effective use of the spectrum. At the same time, in remote areas, there are fewer ground BSs. Since aerial network equipment or space network equipment is far from the ground, this spatial isolation makes it possible to share the spectrum of ground networks without causing serious inter-network interference [8].

There are some surveys on spectrum sharing between aerial/space networks and ground networks, but they mainly focus on a certain research aspect of spectrum sharing. The comparison between different surveys is shown in Table 1. In previous studies, Refs. [9,10] summarized the current spectrum-sensing methods and discussed the advantages and disadvantages of these methods. Although the summary of spectrum sensing technology is relatively comprehensive, these surveys do not have specific application scenarios. Due to the great difference between aerial/space networks and ground networks, spectrum sharing technology needs to be analyzed in specific scenarios. The authors in [11,12] mainly provided a comprehensive description of spectrum sensing, spectrum decision making, spectrum sharing, spectrum mobility and other functions. However, these articles lack a summary of reducing interference between networks. In [13], Jasim et al. conducted a comprehensive investigation of the spectrum management of UAVs and summarized the scenarios and research of spectrum sharing between UAVs and other networks. However, there is little research summarizing spectrum sharing between UAV networks and ground networks in the article, and the performance optimization and resource allocation of spectrum sharing between two networks are not considered. In [14], Hoyhtya et al. summarized the research on spectrum sharing between satellite links and ground or satellite systems in four scenarios and gave the actual use cases and the latest technology. However, the article only summarizes the research on spectrum sharing based on a spectrum database, which is not comprehensive. In [15], the resource allocation in integrated satellite and ground networks was analyzed to optimize energy efficiency and spectrum utilization efficiency. The article summarizes the research on spectrum reuse of satellite networks and ground networks from the perspective of resource allocation, but the scenarios analyzed are limited. With the application of spectrum sharing in more complex scenarios in the future, the summary in the article cannot provide reference in new scenarios.

In this article, our main purpose is to summarize the research on spectrum sharing between aerial/space networks and ground networks. Although there are many studies on spectrum sharing, most of them only focus on ground networks. Because aerial networks and space networks are deployed in high altitude, there are still some challenges in applying spectrum sharing technology to these networks. At present, some surveys only outline the spectrum sharing methods or resource allocation methods in this scenario. Few surveys comprehensively summarize the research on spectrum sharing between aerial/space networks and ground networks. Compared with the previous literature, our article summarizes the scenarios, modes and key technologies of spectrum sharing between aerial/space networks and ground networks. The contributions of this article are as follows:An overview of aerial networks, space networks and space–air–ground integrated networks is presented, and some common application scenarios are summarized.The spectrum utilization rules of aerial networks and space networks are introduced respectively, and the compatibility research of current network coexistence is summarized.The modes of spectrum sharing are divided according to different standards, and common modes are introduced and compared in detail.The key technologies of spectrum sharing between aerial/space networks and ground networks are summarized, including spectrum sensing and interference management.

As shown in Figure 1, the structure of this article is organized as follows. In Section 2, the introduction and application scenarios of aerial networks, space networks and space–air–ground networks are introduced. In Section 3, aerial networks are divided into low altitude platform (LAP) networks and high altitude platform (HAP) networks, and the spectrum utilization rules of aerial networks and space networks are summarized. In Section 4, The spectrum sharing modes between aerial/space networks and ground networks are introduced in detail. In Section 5, common key technologies of spectrum sharing are summarized in detail, mainly including spectrum sensing and interference management technology. In Section 6, the challenges of spectrum sharing between aerial/space networks and ground networks are analyzed. Finally, in Section 7, we conclude the article.

## 2. Scenario Analysis

As the supplements of ground networks, aerial networks and space networks play important roles in fields such as post-disaster reconstruction and traffic offload. In addition, to achieve the global seamless coverage of wireless communications, 6G puts forward the space–air–ground integrated network [16]. The collaboration between aerial networks, space networks and ground networks has been studied extensively. This section mainly describes aerial networks, space networks, space–air–ground integrated networks and corresponding application scenarios.

### 2.1. Summary of Aerial and Space Networks

#### 2.1.1. Aerial Networks

Aerial networks are distributed between ground networks and space networks, mainly at the altitude of 30 km above the ground [5]. They are mainly composed of aircrafts, such as UAVs, aviation aircrafts and so on. The aerial BSs can act as an extension of ground networks to provide a wider communication range for ground users. In addition, they act as relays for communication between ground and space networks. Since there is a certain distance in space between aerial networks and ground networks, there are opportunities for spectrum sharing. By sharing the spectrum with ground networks, communication capacity and spectrum utilization can be improved. Based on their height, aerial network platforms can be classified as HAP and LAP, and they are introduced as follows.

LAP Networks

LAPs, mainly UAVs and aviation aircrafts, are distributed in the air below 10 km from the ground [17]. In recent years, the application fields of LAPs have been continuously expanded, showing their great potential in post-disaster rescue and the expansion of network coverage. Since UAVs mainly access unlicensed bands, with the continuous increase in wireless devices, it is urgent to find new spectrum resources to improve the network capacity of UAVs meeting higher communication requirements [8]. As an effective method to improve spectrum utilization, spectrum sharing has attracted wide attention. As shown in Figure 2, UAVs can communicate with each other or with ground users to provide services. The UAV-to-UAV communication and the communication between UAVs and ground users can share the spectrum of ground networks.

HAP Networks

HAP, which can be regarded as a complementary network of ground networks or space networks, is mainly distributed at a distance of 17–22 km from the ground. Due to the advantages of wide coverage, low delay and flexible deployment, HAPs have attracted widespread attention in recent years [17,18,19]. Compared with ground networks, the communication range covered by HAPs is wider. As a result, HAPs can be used as supplements to ground networks. Compared with space networks, HAPs have a lower delay and can be used as a relay for ground users to communicate with users from the space networks. As the ITU has allocated a spectrum for HAPs, the spectrum sharing between HAPs and other networks has attracted wide attention. For example, [20,21] studied the spectrum sharing between the HAP network and ground WiMAX system. The performance of the two systems was analyzed. The results show that the HAP network can coexist with the ground WiMAX system by using spectral etiquette. The spectrum sharing between the HAP network and 3G mobile network in different environments (suburban macrocellular environment, urban macrocellular environment and urban microcellular environment) under natural disaster scenarios was studied, and the study showed that the HAP network provides higher capacity than the ground network [22]. Similar to the spectrum sharing between LAP networks and ground networks, the spectrum sharing between HAP networks and ground networks is shown in Figure 3.

#### 2.1.2. Space Networks

Space networks mainly consist of satellites. Satellites can be classified into three categories according to their altitude: geosynchronous orbit (GEO) satellites, medium-Earth-orbit (MEO) satellites and low-Earth-orbit (LEO) satellites [23]. Due to the advantages of wide coverage, high capacity and not being affected by terrain conditions, space networks can be used as supplements to the ground networks in broadcasting, emergency communication and remote area communication [24]. Although satellite communications play an important role in providing connectivity and coverage, satellites are limited due to masking in dense areas. Therefore, considering the advantages of ground networks and the weaknesses of satellite networks, a hybrid satellite–terrestrial network is proposed [25,26]. At the same time, spectrum sharing between satellites and ground networks is helpful in solving the problem of a crowded spectrum [27]. Interference between space networks and ground networks is an inevitable problem in spectrum sharing. Guidolin et al. [28] studied spectrum sharing between fixed satellite service (FSS) and cellular networks in the millimeter wave (mmWave) and analyzed the feasibility of spectrum coexistence. Gao et al. [29] proposed a satellite–ground-integrated network, where the ground network shared the spectrum of the satellite network, and analyzed the mutual interference between the two networks and measures to reduce the interference (such as power control). Wang et al. [30] analyzed the interference of International Mobile Telecommunications (IMT) to FSS when IMT and FSS coexisted at 28 GHz, and the experiment showed that spectrum sharing between these two services is feasible.

#### 2.1.3. Space–Air–Ground Integrated Networks

At present, the communication range of human beings is mainly concentrated in the space of several kilometers on the land surface. With the deployment and commercialization of the fifth-generation (5G) mobile communication system, more and more communication devices are distributed in emerging space, aerial and underwater networks. The coverage of these areas cannot be realized only by relying on the current ground cellular networks. 6G has put forward the vision of realizing global, all terrain and all space three-dimensional coverage. Therefore, it is necessary to integrate aerial networks, emerging space networks and underwater networks on the basis of ground networks to provide anytime and anywhere network access [16]. At present, the development of ground networks is relatively mature, and there are still uncertainties about whether wireless communication in underwater networks can be used for 6G. Therefore, in the future, the space–air–ground integrated networks composed of ground networks, aerial networks and space networks will become the focus of research. The structure of the space–air–ground integrated network is shown in Figure 4. In the space–air–ground integrated network, the ground communication service is not only provided by ground BSs but also by aerial networks and space networks, thereby increasing the range of communication and improving the network capacity.

### 2.2. Application Scenarios

Compared with ground networks, the deployment of aerial and space networks is more flexible and less susceptible to terrain and natural disasters. As a result, they are extensively used for emergency communication and wide-scale coverage communication.

In terms of emergency communication, ground networks are susceptible to natural disasters and tend to become paralyzed, making post-disaster rescue more difficult. As a result of the breakdown of ground networks, the spectrum allocated to the ground networks is in an idle state. At this point, aerial networks or space networks are able to utilize the idle spectrum so that they can construct communication networks quickly and provide services to disaster-stricken areas [31].

In terms of large-scale coverage, remote areas are vast and sparsely populated, and the cost of establishing ground BSs is high. Aerial networks and space networks can provide communication services for remote areas due to their advantage of wide coverage. In addition, users at the edge of the cells experience severe path loss, which leads to poor communication quality. The use of aerial networks and space networks can also assist ground networks in offloading traffic and improving communication performance [32]. In this scenario, aerial networks and space networks are far from the ground networks and have less interference. Therefore, the spectrum of ground networks can be shared under the premise of reasonably controlling transmitter power and other parameters of aerial networks and space networks.

In addition to the above application scenarios, some ground services can also be offloaded to aerial networks and space networks to reduce the burden on the ground networks. Excessive use of network equipment in areas with dense populations, such as concerts, will result in the overcrowding of the spectrum, which in turn will lead to the deterioration of communication services. In this case, aerial and space networks can be used to offload ground services [33].

Although both aerial networks and space networks can be applied in the above scenarios, due to factors such as coverage and cost, different networks will be selected to provide communication services in different scenarios. For example, in the Wenchuan earthquake that happened on 12 May 2008, China used maritime communication satellites to provide communication services, which laid a foundation for the success of the relief. In September 2017, Project Loon launched by Google used high-altitude balloons to provide communication in the Caribbean after Hurricane Maria hit the region [4]. Comparatively speaking, space networks are deployed at the highest altitude among the three networks, which can achieve global coverage but have higher propagation delays and propagation losses. Aerial networks have lower propagation delays and experience lower propagation loss than space networks, so they exhibit both the characteristics of space and ground networks. However, problems such as limited capacity and unstable links should be fully considered during deployment [23].

## 3. Current Spectrum Utilization Rules

In order to support the development of aerial networks and space networks, international organizations and countries have allocated fixed frequency bands and designated corresponding rules for the networks. However, there are usually multiple networks coexisting in the spectrum. This section gives a detailed summary of the spectrum utilization of the current aerial/space networks and summarizes the compatibility research on the coexistence of multiple networks in the same frequency band.

### 3.1. Spectrum Utilization of LAP Networks

UAVs are the typical representatives of LAPs, and they mainly provide communication services by accessing unlicensed spectrum. The unlicensed frequency band mainly refers to the 2.4 GHz band and 5 GHz, which is generally used in the industrial, scientific and medical (ISM) fields [34]. In the unlicensed frequency bands, massive wireless devices (such as Bluetooth and WiFi) coexist. Therefore, it is significant to consider the problem of interference management when multiple devices coexist. In the future, the scale of wireless devices and UAVs that access this frequency bands will continue to increase, and the spectrum will become increasingly crowded.

Based on spectrum sharing, the ITU has studied the compatibility of UAV services with other services in order to allocate spectrum for UAV communications without interfering with other services coexisting in the spectrum. For example, Rep.ITU-R M.2229 reported a study on the compatibility of UAV control and non-payload communication (CNPC) links in 15.4–15.5 GHz with UAV positioning, radio navigation and radio astronomy services. The report pointed out that, if proper operation techniques are not adopted, it is difficult for UAV systems and radiolocation services to be compatible on this frequency band [35]. Rep.ITU-R M.2205 studied the use of 960–1164 MHz for ground line-of-sight (LoS) communication of UAV services, mainly referring to LoS communication between the UAV and the ground control station [36]. Although this frequency band has been used by many navigation systems, some of its subfrequency bands (such as 960–976 MHz and 1151–1156 MHz) have not been used and can be used by UAVs to complete the auxiliary communication of ground networks. The details about the ITU studies on the compatibility of UAVs with other systems are summarized in Table 2. In addition, the Ministry of Industry and Information Technology of China has begun allocating dedicated frequency bands for civilian UAV systems in accordance with the “Regulations on Radio Frequency Allocation of the People’s Republic of China” and national spectrum utilization. The results of spectrum allocation are shown in Table 3 [37]. Among them, the services of UAVs coexist with other services in the 840.5–845 MHz band, such as radio frequency identification technology. The coexistence of UAVs’ services and wireless LAN technology also exists at 2408–2440 MHz. In addition, 841–845 MHz can also be used for uplink remote control and downlink telemetry links for aircraft systems in time-division mode. The 1430–1438 MHz band is used for video transmission of police UAVs and helicopters, while 1438–1411MHz is used for other UAVs.

### 3.2. Spectrum Utilization of HAP Networks

Since 1997, ITU-R has been conducting research on HAP. As one of the most important ITU meetings, the World Radio Conference (WRC) has continuously improved the existing wireless communication regulations and spectrum division through proposals from various countries to accommodate the development of new communication technologies and communication services.

The ”Radio Regulations” adopted before WRC-19 allocated three frequency bands for HAP’s FS, namely 47/48 GHz, 28/31 GHz and 6 GHz. Among them, 47/48 GHz is a global frequency band where HAP shares spectrum with GEO satellites and ground services on a harmless and unprotected basis [18]. The 28/31 GHz is designated to be used in Region 1 (R1, including Arab countries, Africa, Europe and Commonwealth of Independent States (CIS) countries) and some countries in Region 3 (R3, including Asia and Pacific). The 6 GHz is used for HAP gateway communication and can only be used in five countries, including Australia. WRC-19 studied additional spectrum requirements of high-altitude platform system (HAPS) gateways and fixed terminal links, the availability of 38–39.5 GHz in the world and the availability of 21.4–22 GHz and 24.25–27.5 GHz in Region 2 (R2). The spectrum division is shown in Figure 5, and the networks coexisting in the spectrum are shown Table 4 [38].

In WRC-19, the proposal from China stated that the current frequency bands allocated to the HAP system are not fully utilized, and the existing spectrum can be used to meet the demand [38]. The underutilized spectrum can also be shared with other networks to improve spectrum utilization. In addition, when the current spectrum allocation of HAP fails to meet the need, it can also access the authorized spectrum of ground networks or satellite networks to realize spectrum sharing. Therefore, in the current spectrum division, there exists the coexistence of multiple systems. For example, in the 5850–7075 MHz, the ground systems, microwave access, and HAP networks coexist [39]. In R2, the 24.25–27.5 GHz and 38–39.5 GHz bands are not only allocated to HAP but also the uplink and downlink of FSS [40]. In addition, the spectrum allocated to HAP also overlaps with the candidate frequency band of 5G, so it is necessary to study the coexistence between the HAP system and other systems.

### 3.3. Spectrum Utilization of Space Networks

At present, the most commonly used frequency bands for satellite communication services include the L, S, C, Ku and Ka bands, shown in Figure 6 and Table 5. For FSS, the bands used for communication is mainly concentrated in the C band and Ku band. The transmission of the C band is relatively stable, but it is easy to cause interference to ground systems. The bandwidth of the Ku band is large, but it is seriously affected by rain decline. Therefore, it is not as stable as the C band [41]. Mobile satellite services (MSS) mainly use the L and S bands to provide services [41]. With the increasing demand for spectrum, the Ka band has been developed to meet the growing satellite communication services. However, there are not only satellite communication services in these bands. For example, for ground service IMT-2000, an extended C band is allocated to provide communication services [42]. In addition, the demand for 5G high bandwidth urgently requires the expansion of new bands. mmWave communication has become a key technology of 5G. For mmWave, 17–30 GHz as a candidate frequency band also brings interference to satellite communication [43]. Therefore, it is necessary to investigate the spectrum sharing between space networks and ground networks.

## 4. Spectrum Sharing Modes

CR technology is a key technology to realizing spectrum sharing. With CR technology, networks that need spectrum sharing can perform spectrum sensing and detect spectrum holes, so as to provide communication services using idle spectrum [46]. In CR technology, the users who participate in spectrum sharing are usually divided into primary users (PUs) and secondary users (SUs). PUs are those who are willing to share spectrum with other access users and have the highest priority to access spectrum [46]. SUs share the spectrum of PUs.

According to different standards, spectrum sharing modes can be classified into different types. As shown in Figure 7, according to the spectrum type, spectrum sharing can be divided into authorized spectrum sharing and unlicensed spectrum sharing. PUs in authorized spectrum sharing have the highest priority to use the spectrum, while in unlicensed spectrum sharing, all users use the spectrum equally. According to the network structure, spectrum sharing can be divided into centralized spectrum sharing and distributed spectrum sharing. Centralized spectrum sharing means that the resource allocation and spectrum access process are determined by the central BS. The advantage of this mode is that, in the spectrum allocation process, the central authority can make priority decisions according to the application type. However, the overhead incurred during information transmission is one of the key constraints [47]. Distributed spectrum sharing means each node in the network determines the spectrum allocation and access process according to its application. The advantage of this mode is that it reduces the signal processing process. However, it increases the computational complexity of each SU [47]. According to the access mode of SUs, spectrum sharing modes are divided into interweave underlay and overlay. In the current literature on spectrum sharing between aerial/space networks and ground networks, the common spectrum sharing modes are interweave, underlay and overlay. The main differences between the three modes are shown in Table 6.

### 4.1. Interweave

In this mode, SUs perform spectrum sensing to determine whether PUs exist. When PUs are using the spectrum for communication, SUs are prohibited from using the spectrum. Since PUs and SUs do not transmit signals at the same time, there is no interference problem [48]. However, the use of spectrum by SUs is carried out on the premise that PUs do not use the spectrum. When PUs need to access the spectrum, SUs must release the occupied spectrum, which will cause communication interruption of SUs. Since PUs and SUs cannot use spectrum at the same time, this mode is suitable for scenarios where PUs spend less time on spectrum.

The key issue in this mode is the detection of the available spectrum. The accuracy of spectrum detection severely affects the performance of two networks during spectrum sharing. Traditional spectrum detection technologies are mainly energy detection, matched filter and cyclostationary feature detection. These technologies are introduced in detail in Section 5. However, when the signals of PUs are relatively weak, the traditional spectrum detection methods cannot detect the existence of PUs. Cooperative spectrum sensing technology can be used to enable multiple LAP network users to sense the presence of ground network users, thereby improving the performance of spectrum sensing [49].

### 4.2. Underlay

In this mode, PUs and SUs simultaneously transmit data, in which the transmitter power of SUs should be smaller than a certain threshold to avoid interference to PUs. This mode is one of the most commonly used in the current spectrum sharing modes. The studies in [31,33,50,51] are all carried out under this mode, and the network performance and optimization measures are analyzed. In this mode, PUs and SUs access the spectrum simultaneously, which significantly improves the spectrum utilization. However, due to the coexistence of different networks, there will be interference between PUs and SUs, which affects the communication service quality of the two networks. Since SUs needs to reduce the interference to PUs, the transmitter power is limited. Therefore, this mode can only be used for short-range communication [52].

According to the above analysis, in this mode, it is significant to reduce the interference of SUs to PUs [48]. Meanwhile, SUs still need to pay attention to spectrum switching. Santana et al. [53] pointed out that LAP is characterized by high mobility. When LAP network users find that PUs are transmitting data, the most effective way to avoid interference is to stop transmission and use other standby spectrum for transmission, which involves spectrum switching. The delay of spectrum switching can be reduced by combining pure proactive and reactive strategies. The pure proactive strategy is to anticipate the arrival of PUs while sensing the spectrum environment based on channel traffic, while the reactive strategy is to perform spectrum switching when PUs arrive. Relatively speaking, the switching delay of the pure proactive strategy is shorter, but it has errors. When the prediction is not accurate, it may cause unnecessary switching. Although the reactive strategies do not have such errors, they have higher switching delays [53]. When space networks share spectrum with ground networks, there are generally two ways to reduce the interference: setting up protection areas and using beamforming. Zhang et al. [45] proposed a universal spectrum sharing framework between satellite network and ground network by analyzing the interference caused by ground cellular network and non-geostationary systems to geostationary systems and obtained a closed-form expression for the interference and radius of the protected area of geostationary systems. In addition, the directional transmission of signals by controlling the antenna beam is also a common method to reduce interference. Sharma et al. [54] proposed a transmitting beamforming method of the BSs in underlay mode. Tani et al. [55] proposed a beamforming method for the coexistence of remote sensing satellite and ground networks, in which the satellite adaptively controlled the visual axis of the transmitting antenna. The objective of these two articles is to maximize the signal-to-interference-noise ratio (SINR) of ground stations of satellite networks, so as to reduce the interference to ground networks.

### 4.3. Overlay

In this mode, PUs and SUs can use the same spectrum at the same time. SUs have the prior knowledge of PUs’ transmission and they can adopt two strategies. One is to use all of their power to transmit data [49], and the other is to assist PUs in transmitting data while transmitting data [48]. Current research mainly concentrates on the second strategy. SUs can make use of the priori information of PUs and adopt relevant encoding technologies (such as rate-splitting, cooperation and superposition coding) to offset the interference to PUs, so as to ensure the communication service quality of PUs remains unchanged or even improves and, meanwhile, improve the spectrum utilization rate [49,56]. However, in this mode, it is still impossible to avoid the interference of SUs with PUs. Since SUs need to obtain the prior information of PUs, this mode is suitable for scenarios where PUs can cooperate with SUs [57].

In this mode, measures need to be taken to protect PUs from SUs’ interference. Tabakovic et al. [58] gave a detailed description of CR technology based on overlay mode and focused on the interference constraints under this mode; this literature proposed two approaches (Block Edge Mask approach and Aggregate PFD approach) to define spectrum usage rights (SUR). For spectrum sharing between LAP networks and ground networks, one of the challenges is power supply. When the UAV provides communication service, the longer the flight distance, the more energy it consumes. In order to balance the relationship between energy efficiency and flight distance, Hu et al. [59] reduced energy consumption by optimizing the UAV’s position and power under spectrum sharing.

In the research on spectrum sharing between ground networks and space networks, Non-Orthogonal Multiple Access (NOMA) is often used to reduce inter-network interference. NOMA allows multiple users to share time, space and frequency. The principle of NOMA is to use non-orthogonal transmission at the transmitter and actively lead into interference information. The receiver realizes correct demodulation through serial interference cancellation technology, thereby improving spectral efficiency. For example, NOMA allows using the same spectrum to meet the needs of different users in different areas for different services, so as to improve the system capacity [60]. Therefore, the application of NOMA to the spectrum sharing of hybrid satellite–ground networks has attracted extensive attention. Zhang et al. [61] proposed a collaborative spectrum sharing based on NOMA and divided the whole signal transmission into two stages. First, the satellite broadcasted the cognitive relay signals to the ground. Then, the ground network relay allocated the NOMA power according to the instantaneous channel conditions, so that the weak users can obtain more transmitter power than the strong users, improving the fairness of the distribution. Then, the mixed signals are sent to PUs and SUs. Le et al. [62] derived the expression of the outage probability and throughput under the NOMA scheme, proving that NOMA is superior to the Orthogonal Multiple Access (OMA) in terms of the outage probability and throughput. Based on the analysis in [62], Refs. [63,64] also gave an asymptotic expression of outage probability, the system performance of which is easier to analyze under high signal-to-noise ratio (SNR).

## 5. Key Technologies of Spectrum Sharing

Combined with CR technology, as shown in Figure 8, the following four steps are required for spectrum sharing: spectrum sensing, spectrum decision, spectrum sharing and spectrum mobility [48]. Spectrum sensing is the detection of an available spectrum. At present, technologies such as energy detection, matched filter and cyclostationary feature detection have been proposed to detect the spectrum usage of PUs. During the detection process, SUs may perceive multiple available spectra. At this time, it is necessary to make a decision to access the most suitable spectrum. The spectrum decision is to select the most appropriate spectrum for communication after fully understanding the current radio environment. There are many criteria for selecting the optimal spectrum, such as the principle of optimal channel quality and the principle of least channel interference. After accessing the appropriate spectrum, SUs can use the spectrum for data transmission and other services.

In order to realize spectrum sharing between aerial/space networks and ground networks, many key technologies have been proposed. For example, in order to better identify the idle spectrum, spectrum sensing technology is proposed; in order to improve the communication performance when multiple networks coexist, the joint optimization algorithm is proposed; in order to reduce the interference between networks, power control, resource allocation and beamforming technologies are applied to the spectrum sharing between aerial/space networks and ground networks. This section mainly summarizes the current key technologies.

### 5.1. Spectrum Sensing

To identify idle spectrum, SUs need to fully understand the current radio environment. As one of the significant technologies of spectrum sharing, spectrum sensing is an important prerequisite for spectrum allocation and sharing. The traditional detection methods include energy detection, matched filter and cyclostationary feature detection, and their main features are summarized in Table 7. Collaborative spectrum sensing and compressed spectrum sensing algorithms are also proposed to improve sensing accuracy and efficiency. With the progress of computing technologies, machine learning has aroused great interest and recently has shown amazing potential in solving problems related to spectrum sensing [25]. In order to facilitate readers to systematically understand the methods of spectrum sensing. This section mainly analyzes and summarizes the existing research on spectrum sensing.

#### 5.1.1. Energy Detection

Energy detection is an algorithm with low computational cost, which is mainly used in scenarios where the SUs have no prior information of PUs’ signals [10]. Its major principle is that the SUs determine the existence of the PUs according to whether the received signals exceed the set threshold value. The received signal at the ith SU can be model as
(1)$$\begin{array}{cc} H_{1}: & y_{i}\left(n\right)=s_{i}\left(n\right)+u_{i}\left(n\right)\\ H_{0}: & y_{i}\left(n\right)=u_{i}\left(n\right), \end{array}$$
where the hypotheses $H_{0}$ and $H_{1}$ represent the absence and presence of PU, respectively. $s_{i}\left(n\right)$ and $u_{i}\left(n\right)$ are the signal from PU and additive white Gaussian noise (AWGN), respectively. Then, the average power received by ith SU can be calculated as
(2)$$T_{i}\left(y\right)=\frac{1}{N}\sum_{n=1}^{N}{\left|y(n)\right|}^{2},$$
where N=2TW is the number of samples. *T* and *W* are sampling time and bandwidth, respectively. Then, $T_{i}\left(y\right)$ is compared with a threshold γ to decide whether the PU exists [65,66].
(3)$$T_{i}(y)\overset{H1}{\underset{H0}{\overset{>}{<}}}\gamma$$

In the research on spectrum sharing between LAP networks and ground networks, Shang et al. [34,67] adopted this method to perform spatial spectrum sensing in the sphere space with a UAV as the center of the circle. The UAV judged whether to access the spectrum according to the comparison between the received energy and the preset energy threshold, and the machine learning method was used to verify the advantages of spectrum access based on spatial spectrum sensing. However, the energy detection method also has some disadvantages. As the noise power changes with time, when the threshold is fixed, the noise power may exceed the predefined threshold, which causes an error in the decision. Especially in fading channels, the performance of energy detection is seriously degraded [68].

Based on the above analysis, the key problem in energy detection is whether the status of PUs can be correctly judged. The main indicators to measure the detection performance are the probability of missing detection and the probability of false alarms. The probability of missing detection $P_{m}$ is the probability of detecting the absence of the PUs when they actually exist, it can be expressed as
(4)$$P_{m}=\mathrm{Pr}\left(H_{0}|H_{1}\right),$$
where $\mathrm{Pr}\left(*\right)$ means the probability of event *. The false alarm probability $P_{\mathrm{fa}}$ is the probability of falsely detecting the PU when the PU is actually absent in the scanned frequency band [34]; it can be expressed as
(5)$$P_{\mathrm{fa}}=\mathrm{Pr}\left(H_{1}|H_{0}\right).$$

In [69], Chen et al. studied the space–time spectrum sharing between UAVs and PUs on the ground, dividing the space into a restricted area and an unrestricted area from bottom to top. In the restricted area, UAVs can only access the spectrum when PUs are not using the spectrum. In the unrestricted region, UAVs can directly share the spectrum with PUs because the distance between UAVs and PUs is far. According to the spectral sensing based on energy detection, ref. [69] calculated the height boundaries of restricted and unrestricted areas under the conditions that the probability of missing detection, the probability of false alarm and the disturbance of UAVs to PUs are all less than the set threshold.

In addition, signal acquisition is also a major research problem. Spectrum sensing in an ultra-wideband (UWB) spectrum is usually limited by the sampling capacity. In this context, the compressed sensing (CS) method has been put forward to reduce the difficulty of spectrum sensing [70]. Conventional compressed spectrum sensing can be divided into three steps: signal collection, signal reconstruction and spectrum detection. Considering that the performance of the existing spectrum sensing schemes based on the CS is limited by the non-strictly sparse spectrum, Xu et al. [70] proposed the iterative compression filtering method, which can identify the spectrum occupied by PUs without restoring or reconstructing the acquired signal, thus simplifying the spectrum sensing process. The research on compressed spectrum sensing will be detailed in Section 5.1.6.

#### 5.1.2. Cyclostationary Feature Detection

Considering that the spectrum of the modulated PU’s signal has correlation and periodicity, cyclostationary feature detection uses the spectral correlation function (SCF) to verify whether the transmitted signal has periodicity, so that SUs can distinguish the noise from the PU’s signal, which improves the performance of the algorithm in low SNR channels [10]. Assuming that the signal of PU is second-order cyclostationary, the cyclic autocorrelation function is
(6)$$R_{x}^{\alpha }\left(\tau \right)=\frac{1}{T}\int_{-T/2}^{T/2}R_{x}\left(t,\tau \right)e^{-j2\pi \alpha t}dt,$$
where *T* is the cycle period and $R_{x}\left(t,\tau \right)$ is the autocorrelation function of transmitted signal x(t). The spectral correlation function of x(t) is [71,72]
(7)$$S_{x}^{\alpha }\left(f\right)=\int_{-\infty }^{\infty }R_{x}^{\alpha }\left(\tau \right)e^{-j2\pi f\tau }d\tau .$$

For AWGN, $S_{u}^{\alpha }\left(f\right)=0$. Since the signal of PU and noise are uncorrelated, the spectral function of the received signal $y_{i}\left(n\right)$ in (Equation 1) is
(8)$$S_{y}^{\alpha }\left(f\right)=\left\{\begin{array}{cc} H_{1}: & S_{x}^{\alpha }\left(f\right)\\ H_{0}: & 0 \end{array}\right.$$

Therefore, it can be determined whether the PU exists by performing cyclostationary feature detection on the received signal.

Compared with the energy detection method, cyclostationary feature detection has higher accuracy and can distinguish the PU’s signal from the noise well, even in the case of low SNR. However, the most serious disadvantage of this method is that the calculation is complex and the processing time is long [53].

#### 5.1.3. Matched Filter

Compared with energy detection, the matched filter is more accurate, and the detection time is shorter [10]. Liu et al. [68] combined the CS and CR technology and proposed the iterative compressed filtering (ICF) algorithm. The algorithm improved the CS process by using the orthogonal projection method so that the UAV could effectively detect the spectrum holes. However, this algorithm requires a priori knowledge of the signal waveform of PU, so that it can only be applied to the scenarios with the cooperation of the PU.

#### 5.1.4. Spectrum Sensing Based on Database

Due to the higher deployment of satellites, there are some limitations when they use traditional spectrum sensing methods to detect the presence of PUs. The database mode can better coordinate the service quality of PUs and SUs. Therefore, it is preferred for sharing spectrum between satellite networks and other networks. The principle of spectrum sharing based on database is that the SU queries predefined databases to find an idle spectrum. This is also the most common way for satellite networks to share a spectrum with other networks. Hoyhtya et al. [14] proposed a database model, in which operators are asked to provide relevant spectrum information to the database, including geographic data, policies and regulations, existing attributes and availability of frequency channels, as well as historical data to determine the frequency bands, times and regions available for spectrum sharing. Chen et al. [73] studied the spectrum sharing between the satellite network and the cellular network in the mmWave band. The satellite’s ground station stored the broadcast information into a database, and the cellular network’s BSs reported their locations to the database and received feedback from the database, deciding whether to share the satellite network’s spectrum and the power they can transmit. This scheme can significantly reduce the interference of BSs of the cellular network to satellite links. Hyhty et al. [74] developed a spectrum database for the coexistence of the satellite network and the ground network in the Ka band, and the results show that the downlink capacity of the satellite network was significantly improved with the help of the spectrum database. Although there are some studies on spectrum sharing of hybrid satellite–ground networks based on spectrum database, there is still a lack of a practical test platform. Hoyhtya et al. [75] described a verification framework and defined the data and steps needed to build the system, which is helpful in building a database-based spectrum sharing test platform.

#### 5.1.5. Collaborative Spectrum Sensing

Considering the low sensing accuracy of a single user when spectrum sensing, there are some studies that have proposed the collaborative spectrum sensing algorithm to improve sensing accuracy. The main idea of collaborative spectrum sensing is to judge the current spectrum usage by combining the sensing information of multiple users. Hu et al. [76] proposed a distributed collaborative sensing method, that is, multiple users simultaneously perform spectrum sensing, in which satellite terminals were the cooperative distributed sensing nodes. The fusion center combines the status information of the ground network detected by satellite terminals to make decisions on whether to use the spectrum or not. However, in the traditional time-based collaborative spectrum sensing algorithm, spectrum sensing is only carried out before SUs’ data transmission. If PUs occupy the channel during the data transmission and are not perceived by SUs, interference will be caused to PUs. Based on the above problems, Jia et al. [77] put forward the collaborative spectrum sensing algorithm based on bandwidth. The bandwidth is divided into two parts. One part is used for the ground station of the satellite network to jointly perceive the presence of PUs and report the detection results to the satellite to make a decision. The other part is used for data transmission. This method can realize real-time detection of the status of PUs in the process of data transmission. Once the existence of PUs is found in the process of data transmission, the channel can be switched immediately.

#### 5.1.6. Compressed Spectrum Sensing

Compressed spectrum sensing methods are mainly used in large bandwidth scenarios. The principle of this method is to sample the sparse signal at a rate much lower than the Nyquist sampling frequency, thereby reducing the complexity of sampling. In the spectrum sharing scenario where the users of satellite networks are PUs, considering the large bandwidth spectrum allocated by the satellite at present, the traditional spectrum detection method has certain limitations in terms of sampling rate. So, compressed spectrum sensing is commonly used [78]. For example, Jia et al. [61] proposed a spectrum sensing method in the hybrid satellite–ground network, in which the energy detection method was used to detect the signals of PUs, and a CS framework was proposed to improve the spectrum utilization by combining CR technology and collaboration technology.

### 5.2. Interference Management

In order to ensure the communication performance of multiple networks in the same frequency band, reducing the interference between networks is one of the problems to be solved at present. In the existing research, the common technologies mainly include power control, beamforming technology and the establishment of protection areas. The specific technical research is described in detail in this section.

#### 5.2.1. Beamforming Technology

Beamforming technology enables different networks to use the same spectrum simultaneously in the same geographical location [14], which significantly improves the utilization of the spectrum. It can reduce the interference between networks by controlling the beam direction [55]. It improves the channel gain in the desired direction by controlling the antenna direction and beamwidth and suppresses the interference to the undesired direction. In previous studies, Yu et al. [79] installed directional antennas on UAVs and ground stations to share spectrum and reduce interference between UAV stystems and ground communication systems using the same spectrum resources. When HAP networks share a spectrum with ground network, its location will shift due to the influence of wind and other factors, which will lead to changes in the service range. Considering this problem, Hoshino et al. [80] proposed a beamforming method based on a cylindrical patch array, independently controlling the horizontal and vertical beam directions to fix the HAP’s coverage area. Zakaria et al. [81] put forward a beamforming method based on the *K*-means algorithm and demonstrated the effectiveness of the algorithm through simulation and comparison with other alternative methods. When the satellite networks share a spectrum with ground networks, Cassiau et al. [44] selected the optimal beam combining the received signal power and the interference power to significantly reduce the outage probability of the ground network. The effect is more obvious in the area with low SNR. Sharma et al. [54] took into account the interference of the ground BSs to the satellite terminals and used beamforming technology to reduce the transmitter power of the ground BSs in the satellite–ground communication direction, thus improving the SNR of the ground terminals and realizing the coexistence of C-band WiMAX and FSS.

However, the traditional beamforming technology only considers the radiation mode controlled in the azimuth direction, which is not applicable in the elevation plane. Considering that the satellites are deployed in the high sky, Sharma et al. [82] proposed to use 3D beamforming in the ground terminals of satellite networks to control the radiation pattern of azimuth angle and elevation angle plane, respectively, and increase the degree of freedom of elevation direction to reduce the interference to the ground network. In the current application of beamforming technology, the full analog beamforming is complex, and the full digital beamforming technology due to a large number of RF links leads to high costs. Based on the above two beamforming technologies, Vazquez et al. [83] proposed a hybrid digital–analog beamforming technology, which combined the advantages of the two technologies to keep the interference below a certain threshold.

#### 5.2.2. Establishment of Protected Areas

By setting a protection area for the primary network and making the secondary network deployed outside the protected area, the interference caused by the secondary network can be greatly reduced. At present, there are many works about protected areas. For example, Oh et al. [84] studied the spectrum sharing between the FSS of geostationary orbit (GSO) and the ground network service in the Ka band. The separation distance between the ground receiving station of the satellite network and the BS of the ground network in the worst case was analyzed. Refs. [44,85] analyzed the spectrum sharing scenario where the satellite network users are PUs and derived the protection radius of PUs to promote the coexistence of the satellite network and the ground network in the mmWave. Khawar et al. [86] analyzed the interference problems, opportunities and challenges of spectrum sharing between small cell networks and satellite networks. Small cell networks are deployed indoors or outdoors and need to report their transmission characteristics to a central database. The central database calculates the radius of the exclusion zone to judge whether small cell networks can operate. Results show that larger protection zones are needed when small cell networks are deployed outdoors.

#### 5.2.3. Resource Allocation and Power Control

Resource allocation and power control are common methods to reduce interference in spectrum sharing. By rationally allocating the existing spectrum resources, spectrum utilization can be maximized. By controlling the transmit power of SUs, the interference to PUs can be controlled within a certain range, so that two networks can simultaneously access the spectrum to provide communication services.

In the studies of spectrum sharing between aerial networks and ground networks, to improve the network capacity of HAP and reduce the interference to ground users, a spectrum sharing method based on carrier aggregation (CA) was proposed in [87] without reducing the resource of the HAP network. That is, the ground network uses an unshared frequency band (principal component carrier, PCC) and a shared frequency band (subcomponent carrier, SCC). Only when the ground network using two frequency bands cannot provide services for ground users can HAP access the spectrum to provide communication services. This method not only improves spectral efficiency but also protects ground users from the interference of HAP downlink. Yang et al. [88] analyzed the network performance when the HAP network and ground network share spectrum in the 3.5 GHz band and verified the feasibility of spectrum sharing when the transmitter power of HAP is controllable. Likitthanasate et al. [89] proposed a multilevel outage probability index to evaluate the network performance when the HAPs share spectrum with the downlink of the terrestrial network in the same coverage area. On this basis, a spectrum etiquette based on a directional antenna was proposed. The spectrum etiquette mainly adjusted the transmitter power of the HAP with interference-to-noise ratio (INR) or carrier-to-interference-plus-noise ratio (CINR) of the ground network users as a reference.

In the hybrid satellite–ground network, the control or allocation of transmitter power can also improve the network performance when spectrum sharing. The current research is shown in Table 8. In time-sensitive applications such as video transmission, power control can optimize performance indicators such as delay-limited capacity and outage capacity, while guaranteeing the communication quality of ground network users [90]. In the coexistence of the ground mobile/fixed communication network (MFCN) with FSS and FS, the power control of the MFCN can also reduce the interference to the ground receiving stations of FSS and FS [91]. In addition, power allocation based on channel estimation can also optimize network performance, such as effective capacity. Vassaki et al. [92] analyzed the power allocation algorithms for perfect channel estimation and imperfect channel estimation to optimize the effective capacity of the ground network while ensuring satellite link communication services. Refs. [60,62,63,64] studied the introduction of NOMA into the hybrid satellite–ground network and conducted power allocation according to the channel conditions of each user to reduce outage probability while ensuring fairness. Power allocation can also be combined with spectrum resource allocation to reduce the interference of the whole network by reducing the total transmitter power [93].

### 5.3. Performance Optimization

In the process of communication, ensuring good network performance to meet the normal or even higher communication requirements is important. Common indicators for evaluating network performance include SNR, SINR, outage probability, network capacity, etc. By analyzing the network performance, the feasibility of spectrum sharing of different networks can be evaluated. Additionally, network performance can be improved by optimizing network parameters. As there is much research on spectrum sharing between LAP networks and ground networks, this section provides a summary of the performance analysis and joint optimization in the case of spectrum sharing between LAP networks and ground networks.

#### 5.3.1. Performance Analysis

The literature on spectrum sharing between LAP networks and ground networks is shown in Table 9. It can be revealed that there are mainly two scenarios for spectrum sharing between LAP networks and ground networks: one is that the communication between UAVs uses the same frequency band as the ground communication networks [31,94,95], and the other is that air-to-ground communication uses the same frequency band as the ground communications networks [33,96,97,98]. The study of the second scenario is more extensive. The two scenarios are described in the following paragraphs.

When UAV-to-UAV communication shares spectrum with ground networks, the literature focuses on the performance analysis when the communication between UAVs shares the spectrum with ground networks. When multiple UAVs are distributed at high altitudes, they can form a wireless mesh network to serve users on the ground. Under this scenario, Wei et al. [31] studied the network performance when the communication between UAVs shared spectrum with the ground network and analyzed the influence of parameters such as UAVs’ density and transmitter power on the network performance, in which the UAVs were modeled as three-dimensional Poisson point processes (PPP). It was found that UAVs have an optimal deployment height to maximize the network capacity. In addition, taking into account the actual channel model, antenna pattern and power control strategy, Azari et al. [94] analyzed the spectrum sharing between UAV communication and the uplink of the cellular network and balanced the communication between UAV network and the ground cellular network by studying power control strategy. Considering application scenarios of UAVs, Guo et al. [95] studied the spectrum sharing between UAVs as aerial relays and ground D2D networks under disaster scenarios. When the ground network is destroyed, aerial BSs composed of UAVs are used to transmit data, thus improving the communication efficiency.

When UAV-to-ground communication share spectrum with ground networks, the literature mainly studies the performance of the networks. Zhang et al. [33,97] established a spectrum sharing model on the basis of stochastic geometry theory and analyzed the network performance. In addition, the optimization problem of spectrum sharing between the two networks was also proposed. In order to maximize network throughput, the deployment density of UAVs was optimized, thus improving spectrum efficiency. Mozaffari et al. [33], respectively, analyzed the network coverage rate and total system rate when static UAVs and dynamic UAVs communicated with ground users and shared the special spectrum with the ground D2D network. Among them, ground users served by UAVs obeyed uniform distribution, and D2D users obeyed PPP. For the scenario of static UAVs, the average coverage of network users can be maximized by adjusting UAVs’ height and ground D2D users’ density. In the scenario of dynamic UAVs, the UAVs can completely cover the whole target area in the shortest time with the minimum transmitter power, so it is necessary to analyze the minimum number of docking points of UAVs to provide service for ground users to the maximum extent. In addition to the above analysis of network performance, there are also studies on channel resource allocation to improve the performance of UAV networks and ground networks. Yoshikawa et al. [98] put forward the concept of the main frequency band and the standby frequency band, in which the radar system exists in the main frequency band. The authors in [98] derived the system interrupt probability when UAVs and the radar system coexist and optimized the number of UAVs using the main frequency band without interfering with the radar system. The analysis of network performance can provide some guidelines for the further study of the scenarios of sharing spectrum between LAP networks and ground networks.

#### 5.3.2. Joint Optimization of UAVs’ Deployment Height, Density and Transmitter Power

By studying the network performance of spectrum sharing, the relevant parameters can be adjusted to optimize the network performance. A UAV has the characteristics of fast movement and flexible deployment and can provide communication service for ground users when it is dynamic or static. Therefore, the deployment height, deployment density, path planning and transmitter power of UAVs all are jointly optimized. In addition, the optimized management and allocation of spectrum resources can improve network performance. Generally speaking, network performance is related to multiple system parameters, so when optimizing network performance, multiple parameters are commonly optimized. The current literature on UAV network performance optimization is shown in Table 10.

The static UAVs refer to the UAVs hovering in the high sky at a fixed height to provide services for ground users, so the optimized parameters are mainly the deployment height of UAVs, transmitter power, etc. Huang et al. [32] jointly optimized the deployment height and transmitter power of a UAV according to UAV deployment height limit, transmitter power and interference temperature constraint to maximize the communication rate of ground users of the UAV network. Additionally, due to limited energy storage, UAVs need to complete data transmission in a limited time. Therefore, reducing energy loss in the communication process as much as possible is one of the significant issues to be solved. Ghazzai et al. proposed a joint optimization solution based on the improved Weber algorithm in [50]. By optimizing the 3D position and power distribution of UAVs, the flight time of UAVs and the communication energy consumption were reduced without decreasing the data transmission rate of PUs. Hu et al. adopted a convex optimization algorithm to boost the sensing performance of a UAV while protecting the transmission of the PU by optimizing the position and power of the UAV. Ref. [59] also maximized transmitted bits of data.

In the scenario where a dynamic UAV communicates with the ground users, the UAV is scheduled to fly from the initial position to the key position. Therefore, its path planning is very important. The current literature focuses on improving network throughput by optimizing UAV’s path planning and power allocation. Power control and path planning of UAVs have been studied in [32,51,99,100]. Specifically, local optimal 3D placement and power control were obtained by using the continuous convex approximation algorithm under the limits of the maximum flight speed and the maximum deployment height of the UAV, as well as the power control based on the interference temperature technology in [32,99]. A D.C. algorithm was used to maximize network throughput by jointly optimizing UAV trajectory and power control [51,100].

Resource allocation is also an effective method to improve network performance. Resource allocation mainly includes the allocation of antenna, bandwidth and UAV density. Che et al. [101] proposed that all UAVs as SUs are equipped with directional antennas of adjustable beamwidth, which can reduce interference and improve the quality of UAVs’ downlink communication. In the article, the ground network and UAV-to-ground downlink were modeled by using stochastic geometry, and the successful coverage probability of the network was analyzed. On the basis of protecting the ground users from interference, the UAV beamwidth and UAV density were optimized to improve the UAV coverage probability. Lyu et al. [102] put forward a hybrid architecture composed of UAV and ground BSs, in which the UAV, as an aerial BS, periodically flew along the edge of the cell to offload traffic for edge users far from the ground BSs. In the article, the minimum throughput of terminals in ground cellular networks was improved through the joint optimization of bandwidth allocation, user partitioning and path planning of the UAV. The authors in [103,104] put forward another spectrum sharing strategy on the basis of [102], namely spectrum reuse, and analyzed the network performance under two spectrum sharing strategies (orthogonal spectrum sharing and spectrum reuse). On this basis, Song et al. [103] maximized the minimum throughput of edge users by jointly optimizing the bandwidth allocation rate, the coverage radius of UAVs and the number of UAVs. In [104], Lu et al. maximized the minimum throughput of all mobile users by jointly optimizing the route, bandwidth allocation and user partition of the UAV. In the case of spectrum reuse, the directional antenna of the UAV and the adaptive transmission of ground BS are used to reduce the interference between the two networks to improve the network performance.

## 6. Future Challenges

Compared with ground networks, aerial networks and space networks are highly deployed and have strong mobility. It is precisely because of these advantages that they can be used as complementary networks to ground networks to provide communication services. However, these characteristics also bring some challenges to the application of spectrum sharing technology in aerial/space networks. The main challenges are as follows:Aerial networks and space networks are generally in a non-stationary state. Due to their flexible mobility, the spectrum environment is always changing, which increases the difficulty of spectrum sharing with ground networks. Therefore, fast spectrum sensing methods are required. In addition, the BSs in aerial networks and space networks have limited energy, which requires the development of low-complexity algorithms to maximize energy efficiency within a limited period.Aerial BSs are easily affected by factors such as airflow. Due to the long distance between aerial BSs and ground users, a small fluctuation may cause a large beam offset, which brings difficulties to the spectrum sharing between aerial networks and ground networks. On the one hand, beam offset cannot ensure the beam alignment of an effective air–ground communication link, which will weaken the useful signals received by ground users and reduce the communication quality. On the other hand, beam offset will increase the probability of users in other coexistence networks receiving interference signals, which will cause serious interference with other coexistence networks. Therefore, it is necessary to study effective beam management technology to reduce the interference between networks during spectrum sharing.Before spectrum sharing, spectrum sensing is required to determine the spectrum to be accessed. Considering the poor sensing performance of a single device, cooperative sensing technology is proposed to improve spectrum sensing accuracy. Cooperative sensing technology mainly makes decisions by combining the sensing information of multiple devices. Due to the wide distribution of aerial/space network devices, the devices are far away from each other and the location is generally in dynamic change. Therefore, how to integrate the sensing information of multiple devices is a problem that needs to be solved at present.

## 7. Conclusions

With the rapid increase in wireless devices and the rapid development of wireless services, the spectrum used by aerial networks and space networks is limited and cannot meet higher communication requirements. As an effective method to alleviate spectrum shortage, spectrum sharing has attracted extensive research. In this article, we summarize related research on spectrum sharing between aerial/space networks and ground networks. First, we introduced the concepts and application scenarios of aerial networks and space networks. Then, we divided aerial networks into LAP networks and HAP networks. The spectrum utilization rules of LAP networks, HAP networks and space networks and the compatibility research with other networks were summarized respectively. Next, we introduced the classification of spectrum sharing modes and compare interweave, underlay and overlay in detail. Afterward, we summarized the key technologies for spectrum sharing between aerial/space networks and ground networks. Finally, we put forward the challenge of spectrum sharing between aerial/space networks and ground networks. This article will provide guidance for spectrum sharing between aerial/space networks and ground networks.

## Figures and Tables

**Figure 1 sensors-23-00342-f001:**
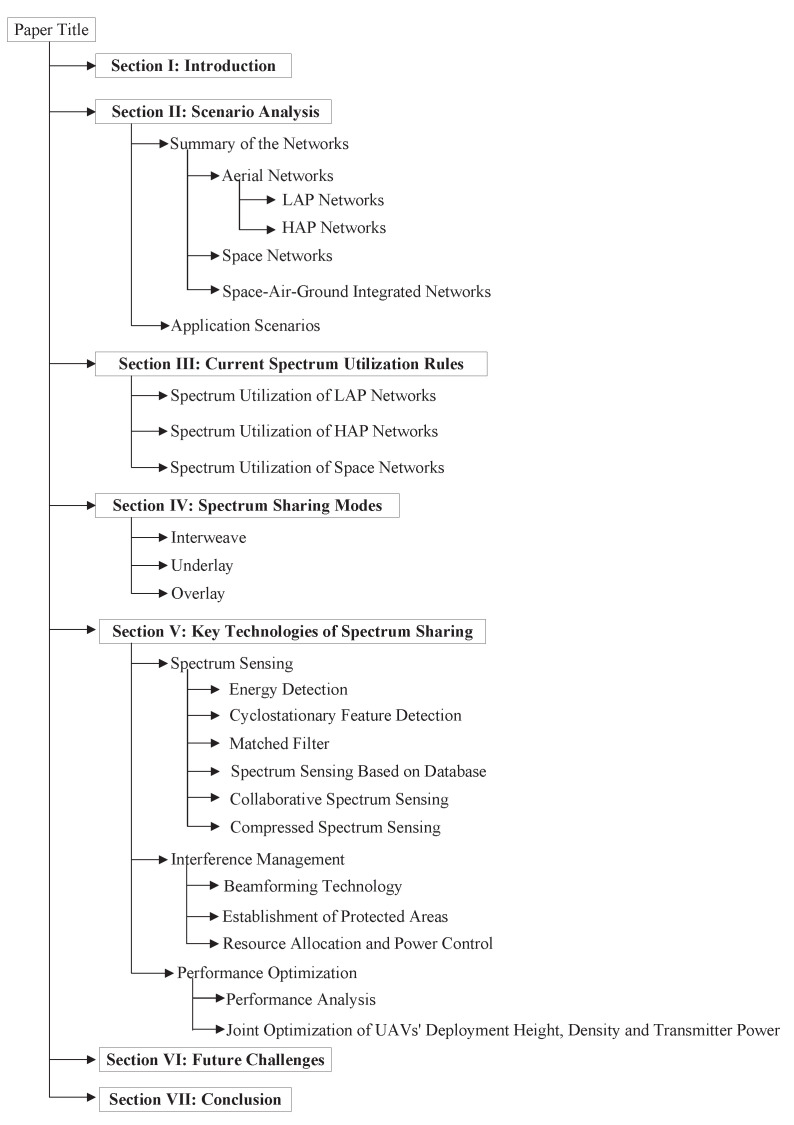
The organization of this article.

**Figure 2 sensors-23-00342-f002:**
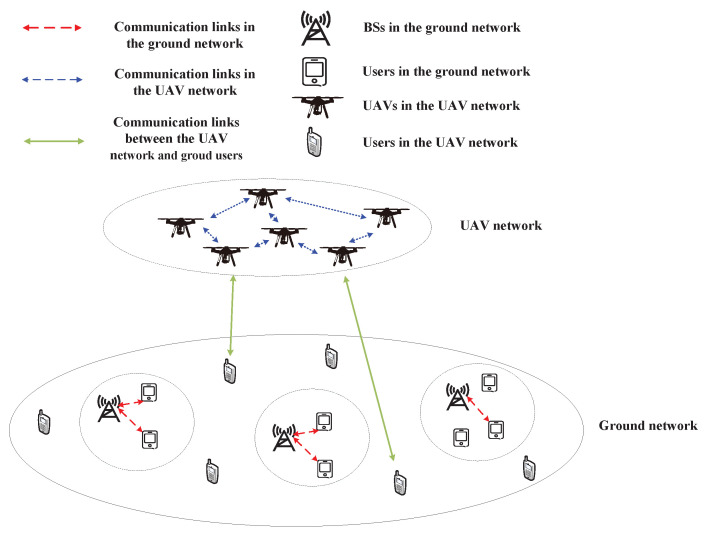
Spectrum sharing between LAP networks and ground networks.

**Figure 3 sensors-23-00342-f003:**
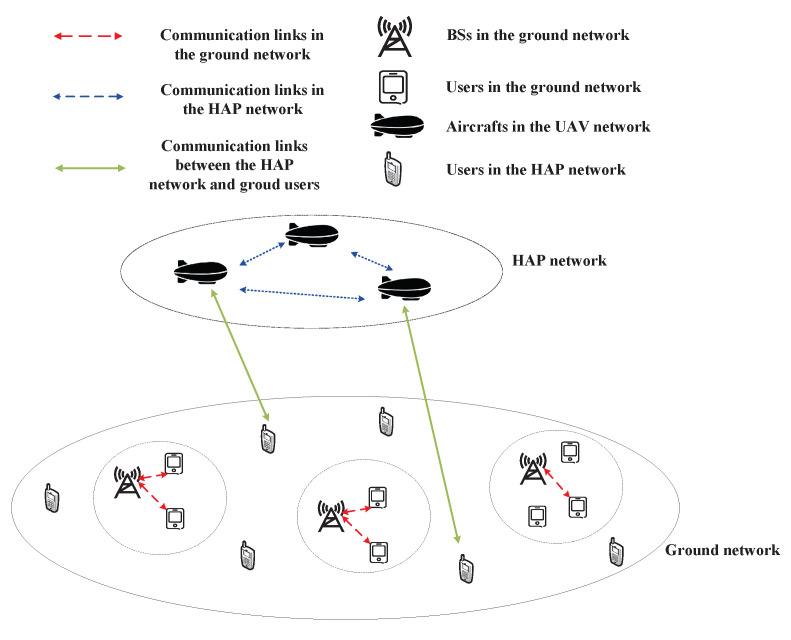
Spectrum sharing between HAP networks and ground networks.

**Figure 4 sensors-23-00342-f004:**
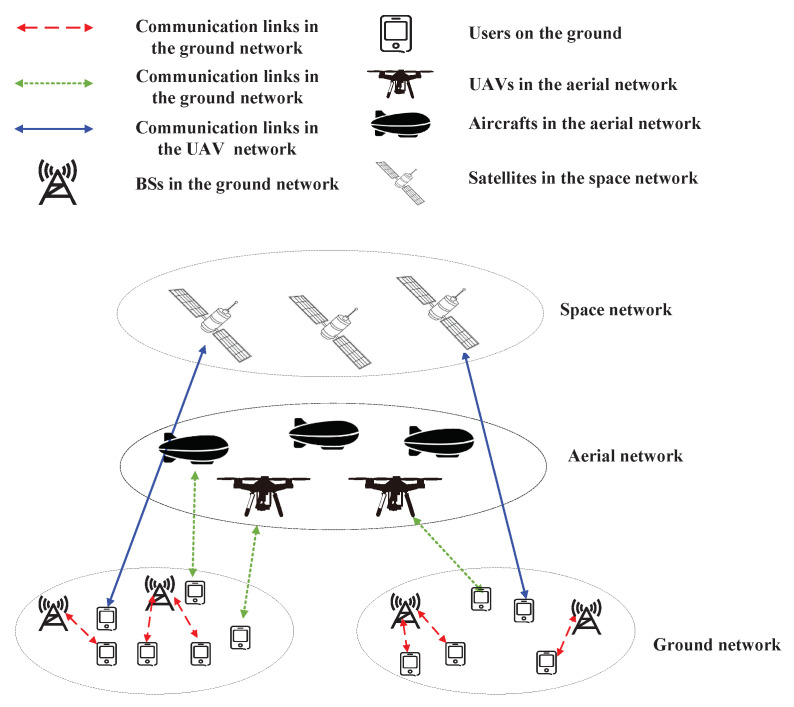
The space–air–ground integrated network.

**Figure 5 sensors-23-00342-f005:**
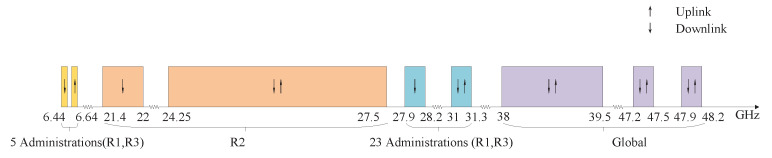
Current spectrum utilization rules of HAP.

**Figure 6 sensors-23-00342-f006:**
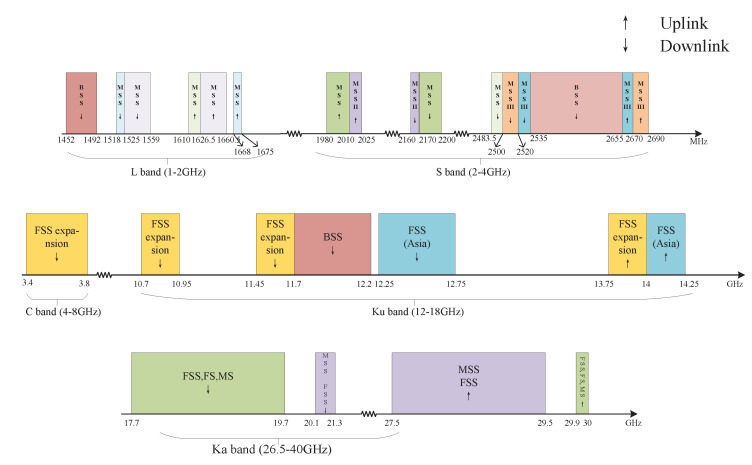
Current spectrum utilization rules of satellites.

**Figure 7 sensors-23-00342-f007:**
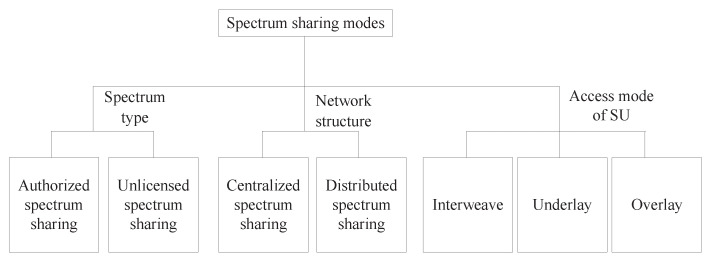
Classification of spectrum sharing methods.

**Figure 8 sensors-23-00342-f008:**
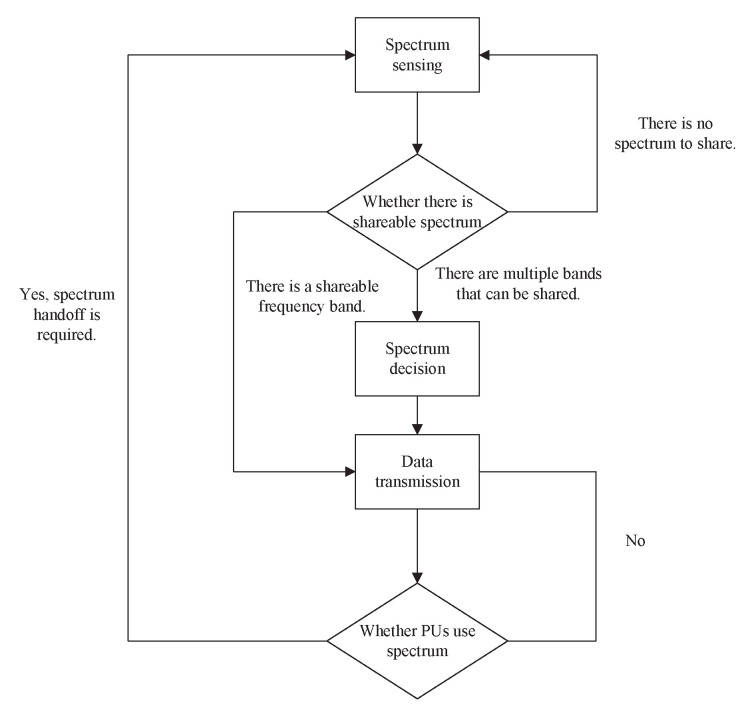
Spectrum sharing process [48].

**Table 1 sensors-23-00342-t001:** Comparison of the related research literature.

References	Spectrum Sharing between Aerial/Space Networks and Ground Networks	Current Spectrum Utilization Rules	Modes of Spectrum Sharing	Key Technologies of Spectrum Sharing
[9]	✕	✕	✕	✓
[10]	✕	✕	✓	✓
[11,12]	✕	✕	✓	✓
[13]	✓	✓	✕	✓
[14]	✓	✕	✓	✕
[15]	✓	✕	✕	✓
This article	✓	✓	✓	✓

**Table 2 sensors-23-00342-t002:** ITU’s studies on the compatibility of UAVs and other services.

Frequency Band	Reference	Research Content	Report Results
960–1164 MHz	Report ITU-R M.2205	Compatibility study of UAV system and aeronautical mobile (route) service (AM(R)S)	Some countries can use a part spectrum of AM(R)S to support part of the demand for UAV systems.
5000–5010 MHz	Report ITU-R M.2236	Compatibility study of UAV system and the radionavigation satellite service (RNSS)	Medium/large UAV system and UAV system control stations are not compatible with RNSS in this frequency band.
5030–5091 MHz	Report ITU-R M.2237	Research on the compatibility of UAV system and microwave landing system (MLS)	In the case of the same occupied bandwidth, it is difficult for the UAV system to be compatible with the MLS system.
5091–5150 MHz	Report ITU-R M.2238	Compatibility analysis of UAV system and ground CNPC links, FSS, AM(R)S and AMS	It is difficult for UAV systems to be compatible with existing services in the frequency band.
13.25–13.4 GHz	Report ITU-R M.2230	Research on the compatibility of UAV systems with aeronautical radionavigation and satellites	This frequency band is not suitable as a candidate frequency band to support UAV system control links in non-isolated airspace.
15.4–15.5 GHz	Report ITU-R M.2229	Compatibility study of UAV system with radiolocation service, aeronautical radionavigation service and radio astronomy service	It is difficult for UAV systems to be compatible with aeronautical radionavigation services and radio astronomy services.
22.5–22.55 GHz and 23.55–23.6 GHz	Report ITU-R M.2230	Compatibility analysis of UAV systems and fixed service (FS)	These frequency bands are not suitable as a candidate frequency band to support UAV system control links in non-isolated airspace.

**Table 3 sensors-23-00342-t003:** Spectrum allocation of UAV (China) [37].

Frequency Band	Application
840.5–845 MHz	Remote control
1430–1444 MHz	Downlink telemetry and data transmission
2408–2440 MHz	Backup frequency band of uplink and downlink telemetry links and information transmission links

**Table 4 sensors-23-00342-t004:** Spectrum allocation of HAP [38].

Frequency Band	Communication Link	Administrations	Existing Network
6440–6520 MHz	Downlink	5 Administrations(R1,R3)	HAP and GEO
6560–6640 MHz	Uplink	5 Administrations(R1,R3)	HAP and GEO
21.4–22 GHz	Downlink	R2	
24.25–27.5 GHz	Uplink and downlink	R2	HAP and uplink of FSS
27.9–28.2 GHz	Downlink	23 Administrations(R1,R3)	
31–31.3 GHz	Uplink and downlink	23 Administrations(R1,R3)	
38–39.5 GHz	Uplink and downlink	Global	HAP and downlink of FSS
47.2–47.5 GHz	Uplink and downlink	Global	
47.9–48.2 GHz	Uplink and downlink	Global	

**Table 5 sensors-23-00342-t005:** Spectrum usage of satellite communication.

Frequency Band	Reference	Uplink	Downlink	Services
L band (1–2 GHz)	[41]	1610.0–1626.5 MHz	2483.5–2500 MHz (S band)	MSS (global)
	[41]	1626.5–1660.5 MHz	1525–1559 MHz	MSS (global)
	[41]	1668–1675 MHz	1518–1525 MHz	MSS (global)
			1452–1492 MHz	Broadcast satellite service (BSS)
S band (2–4 GHz)	[41]	1980–2010 MHz	2170–2200 MHz	MSS (global)
	[44]	2010–2025 MHz	2160–2170 MHz	MSS (Countries in R2)
			2535–2655 MHz	BSS
	[41]	2655–2670 MHz	2520–2535 MHz	MSS (R3)
	[41]	2670–2690 MHz	2500–2520 MHz	MSS (R3)
C band(4–8 GHz)	[14]		3400–3800 MHz	Extended band for FSS
Ku band (12–18 GHz)		14.0–14.25 GHz	12.25–12.75 GHz	FSS (Asia Pacific)
		13.75–14 GHz	10.7–10.95 GHz and 11.45–11.7 GHz	Expansion of FSS in the Ku band
			11.7–12.2 GHz	BSS
Ka band (26.5–40 GHz)	[43]	27.5–30 GHz	17.3–21 GHz	FSS and one of the candidate frequency bands for the deployment of millimeter-wave cellular networks
	[45]	27.5–29.5 GHz	17.7–19.7 GHz	FSS, FS and mobile service
	[41]	29.9–30 GHz	20.1–21.3 GHz	MSS and FSS

**Table 6 sensors-23-00342-t006:** Comparison of interweave, underlay and overlay.

Spectrum Sharing Mode	Data Transmission Characteristics	The Key Problem	Advantage	Disadvantage	Application Scenario
Interweave	SUs access the spectrum when PUs do not use the spectrum.	Detection of the status of spectrum used byPUs.	The communication of PUs is not affected by SUs [48].	When PUs need to access the spectrum, SUs need to release the spectrum immediately, which will cause communicationinterruption.	Lightly occupied spectrumregions.
Underlay	SUs and PUs can provide communication servicessimultaneously.	It is necessary to reduce the interference of SUs toPUs.	PUs and SUs use the spectrum at the same time, which improves the spectrumutilization.	The communication of SUs will interfere with the communication of PUs [48].	Short-range communication.
Overlay	Part of the SUs assists PUs in data transmission, and the rest transmit their own data [48].	It needs to obtain prior knowledge ofPUs.	Due to SUs’ auxiliary communication, the communication quality of PUs remains unchanged [49].	The communication of SUs will interfere with the communication ofPUs.	PU and SU have a high level ofcooperation.

**Table 7 sensors-23-00342-t007:** Comparisons of energy detection, cyclostationary feature detection and matched filter.

Method	Main Idea	Advantages	Disadvantages	Application Scenario
Energy detection	The signal energy from PUs is higher than the set threshold value.	It has the lowest computing cost.	The detection performance in low SNR channels is not ideal [10].	Scenarios where SUs do not have prior information about the signals of PUs.
Cyclostationary feature detection.	Using the spectral correlation of PUs’ signals to analyze the characteristics of the cyclic spectrum in the spectral correlation function of the signals [10].	It can improve the detection performance in low SNR channels.	The calculation is complicated and the processing time is long.	It is mainly used in low SNR scenarios.
Matched filter	It is mainly to analyze the output signal waveforms; when the PUs exist, the output waveforms are the autocorrelation function of PUs’ signals.	The detection accuracy is higher and the detection time is shorter [10].	The prior information of the signal waveforms emitted by PUs is required.	Scenarios where the SUs have prior information of the signal waveforms of PUs.

**Table 8 sensors-23-00342-t008:** Reference related to power control/allocation.

Technology	References	Summary of research
Power control/allocation	[60,62,63,64]	NOMA was introduced into the hybrid satellite–ground network, and power was allocated according to the channel conditions of each user.
	[90]	Using optimization algorithms, the optimal power control scheme for real-time applications in the uplink cognitive satellite–ground network was proposed from the perspective of maximizing the delay-limiting capacity and the interruption capacity.
	[91]	The protection standard was defined with the maximum allowable interference level as the goal. This standard limited the maximum permitted power (EIRP) of the MFCN BS to reduce the interference to the FSS and FSreceiver.
	[92]	The ground BSs adjusted the transmitter power according to the state of the satellite network and the characteristics of the radio channel to protect the satellite link.
	[93]	The transmitter power of satellites and ground BSs was optimized according to the service requirements, so as to minimize the interference between networks.

**Table 9 sensors-23-00342-t009:** Reference related to performance analysis of spectrum sharing.

Scenario	References	The Research Content	The Research Outcome
UAV-to-UAV communications shares spectrum with ground networks	[31]	The spectrum sharing between the UAV mesh network and the ground network is studied, and the influence of UAVs’ density and deployment height on the network performance are analyzed.	With the increase in UAVs’ distribution height, the success probability of communication of ground network users decreases first and then increases, and there is an optimal deployment height of a UAV to maximize the communication capacity of the UAV network.
	[94]	Considering the actual channel model, antenna pattern and power control strategy, the scenario of communication between UAVs sharing spectrum with the uplink of the cellular network is analyzed.	Although the existence of the UAV-to-UAV communications will reduce the communication quality of cellular ground users, such interference is limited. In addition, the performance of a UAV-to-UAV communication link and the uplink of cellular ground users will be improved with the increase in UAVs’ deployment height.
	[95]	The spectrum sharing between UAVs as relays and ground device-to-device (D2D) networks in a disaster scenario is studied.	With the increase in UAVs’ height, the sum rate also increases. Additionally, there exists an optimal UAV height that can maximize the sum rates of downlink users and D2D users.
UAV-to-ground communication shares spectrum with ground networks	[33]	The spectrum sharing between UAV-to-ground communication and D2D network is studied in static and dynamic scenarios, respectively.	According to the density of D2D users, the optimal value of UAV altitude exists, resulting in the maximum system sum-rate and coverage probability. In addition, by enabling the UAV to intelligently move over the target area, the total transmitter power required by the UAV to cover the entire area can be minimized.
	[96,97]	The spectrum sharing between UAV-to-ground communication and the ground cellular network is studied, and the optimal deployment density of UAVs is obtained.	For spectrum sharing between UAV network and cellular network, with the relaxation of the restrictions for UAV network, the independence of the optimal UAV network BSs’ deployment changes from the constraints of the UAV network to the constraints of the cellular network.
	[98]	It provides a channel resource allocation method for the co-existence of UAV-to-ground communication and radar system.	The ratio between the number of UAVs using the main band to the total number of UAVs increases along with the primary exclusive region size.

**Table 10 sensors-23-00342-t010:** Reference related to performance optimization of networks.

Joint Optimization Object	References	The Research Methods	The Research Content
Parameters of static UAV	[32]	Semi-definite relaxation (SDR)	The deployment height and transmitter power of the UAV were optimized jointly to maximize the communication rate.
	[50]	Improved Weber algorithm	By optimizing the three-dimensional (3D) position and power distribution of UAVs, the flight time of UAVs and the communication energy consumption were reduced without compromising the communications between PUs.
	[59]	Convex optimization algorithm	By optimizing the position and power of the UAV, the sensing performance and data transmission rates were improved.
Parameters of dynamic UAV	[32,99]	Alternating optimization and successive convex approximation	The 3D trajectory and transmitter power of the UAV were optimized to maximize the average achievable rate of the flight control system during this period.
	[51,100]	Difference of two convex (D.C.) programming and successive convex approximation	D.C. algorithm was used to maximize network throughput by jointly optimizing the UAV’s trajectory and power control.
Resource allocation	[101]	Stochastic geometry theory	The UAV beamwidth and UAV density were optimized to improve the UAV coverage probability.
	[102]	Alternating optimization	A hybrid architecture composed of a UAV and ground BSs is put forward to improve the minimum throughput of terminals in ground cellular networks by optimizing bandwidth allocation, user partitioning and path planning of the UAV.
	[103,104]	Alternating optimization	Two spectrum sharing strategies are proposed, and related parameters are optimized jointly under different strategies to improve network performance.

## Data Availability

Not applicable.

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
