# Peer review of "Spectrum Sharing in the Sky and Space: A Survey"

_sensors, 2022, doi:10.3390/s23010342_

Round 1
Author Response
We would like to sincerely thank you for handling the manuscript and providing us the opportunity to improve the quality of the manuscript. We have carefully revised our manuscript, taking all the reviewers’ comments and suggestions into consideration. The main changes are as follows
- Related references in recent years are added
- More figures about the analytical results are provided.
- The typos and unclear lines are modified.
Detailed responses and revisions can be found in the file.
Best regards,
Authors

Reviewer 2 Report
The paper is a survey about Spectrum Sharing in space/air/ground networks. I think the subject is very interesting but some topics of the presented paper must be better addressed before I can recommend the publication in Sensors. My main concerns will be listed in the next paragraphs:
1) Paper organization: Figure 1 shows the overall paper organization but some subsections are left out. For example, the types of Cognitive Radio Technologies are presented as Subsections of Section 3.1.2 (Overlay, Underlay and Interweave) but not presented in Figure 1. Next, a similar concept is presented in Section IV and mentioned as Sections. The use of four levels of subsections renders the paper very difficult to follow.
In my opinion the paper can be more useful if section III and IV were merged. Both sections discuss the same topics in different contexts (Space, HAP and LAP) but there is a lot of overlap, which renders the paper very difficult to follow. The repeated topics should be merged and the differences between the techniques should be highlighted.
2) The survey is not up to date. The newer citation is already 1 year old. This severely limits the usefulness of the current version. A newer version should include paper published in 2022.
3) Some figures illustrating the spectrum division should be nice.
4) Spectrum sharing figures Captions should briefly explain the exposed scenario. The reds and yellows chosen are too difficult to differentiate. It would be better to use different line styles to different kind of links.
Author Response
We would like to sincerely thank you for handling the manuscript and providing us the opportunity to improve the quality of the manuscript. We have carefully revised our manuscript, taking all the reviewers’ comments and suggestions into consideration. The main changes are as follows.
- The structure of the article is adjusted, and the same topics are merged into a new section.
- More figures about the analytical results are provided.
- The typos and unclear lines are modified.
Detailed responses and revisions can be found in the file.
Best regards,
Authors

Reviewer 3 Report
This paper summarizes the spectrum-sharing techniques between aerial/space networks and ground networks.
There are some observations as follows,
1. Include a comparative analysis between this paper and other existing surveys/review papers of the same kind.
2. The motivation behind this survey is not clear. Please address this in the introduction section.
3. Correct the caption corresponding to Table 4.
4. Include more details analysis corresponding to the ‘Spectrum sensing’ section. Mathematical models and analysis can be incorporated.
5. There is no such discussion on the research challenges. It is very important for a survey paper.
6. It will be better if the authors include a comparative analysis between the existing spectrum-sharing methods/algorithms keeping in mind diverse applications related to the topic of the paper.
7. Considering Table 5. It will be better if the authors mentioned the significant outcomes rather than only stating what was done in that paper.
8. Please maintain uniformity in declaring the section and sub-section no.
Author Response
We would like to sincerely thank you for handling the manuscript and providing us the opportunity to improve the quality of the manuscript. We have carefully revised our manuscript, taking your comments and suggestions into consideration. The main changes are as follows.
- The comparison with other related surveys/reviews are added.
- Mathematical models and analysis are provided.
- The typos and unclear lines are modified.
Detailed responses and revisions can be found in the file.
Best regards,
Authors

Round 2
Reviewer 2 Report
In my opinion, my main concerns about the paper that were raised in the first round of revision were not properly addressed in the newer version of the manuscript and not discussed in the provided answers (which are only brief when explaining the modifications or the decisions about not accepting my suggestions). For this reason, I am afraid I cannot recommend the paper publication.
In my first round, I stated that paper organization renders the paper difficult to read because there is a lot of overlap in the sections. This issue remains: each section has overlapping topics like spectrum sensing, power control, utilization rules, just to name a few. There is no discussion about the similarities or differences between the topics for different use cases which renders the paper repetitive and difficult to follow.
Next, I stated that the provided bibliography was at least 1 year old. The problem remains. Only two 2022 papers were added and briefly mentioned on the newer version of the manuscript. The answer provided by the authors does not mention which references were added to the newer version of the manuscript, which difficult the reviewing work.
Finally, the newer figures are hard to read due to low quality and are not drawn to scale. This way their contribution to the paper is limited.
Author Response
We would like to sincerely thank you for handling the manuscript and providing us the opportunity to improve the quality of the manuscript. We have carefully revised our manuscript, taking all the reviewers’ comments and suggestions into consideration. The response to comments can be found in the attachment. Thank you for your comments!

Reviewer 3 Report
The authors have addressed all the points raised during last review. But there are some observation,
1.Please check the Keywords: mainly survey; review .....better keyword may be chosen.
2. However, due to the wide distribution range of space-air-ground integrated networks, how to integrate information between multiple devices is also a problem that needs to be solved. Not clear .Please rewrite it.
3. On the one hand, beam alignment cannot be achieved. On the other hand, it will cause interference to other network users during spectrum sharing. Please represent it in a better way.
Author Response

(The authors gave the same response as above.)

Round 3
Reviewer 2 Report
I have carefully read the newer version of the manuscript. The newer version properly addressed my concerns of the first and second review rounds, This way I have no further comments.